



**Investigation of the relationship between drinking water quality and landform classes**
**using fuzzy AHP(case study: south of Firozabad, east of Fars province, Iran)**
**Marzieh Mokarram[1] and Dinesh Sathyamoorthy[2]**
*[1]Marzieh Mokarram(Department of Range and Watershed Management, College of Agriculture and*
*Natural Resources of Darab, Shiraz University, Iran, Email:* m.mokarram@shirazu.ac.ir*)*
*[2]Dinesh Sathyamoorthy (Science & Technology Research Institute for Defence (STRIDE), Ministry of*
*Defence, Malaysia (E-mail: dinesh.sathyamoorthy@stride.gov.my)*
***Corresponding author:*** *Marzieh Mokarram, Tel.: +98-917-8020115; Fax: +987153546476 , Address:*
*Darab, Shiraz university, Iran, Postal Code: 71946-84471, Email: m.mokarram@shirazu.ac.ir*



**Investigation of the relationship between drinking water quality and landform classes**
**using fuzzy AHP(case study: south of Firozabad, east of Fars province, Iran)**
**Abstract**
In this study, fuzzy analytic hierarchy process (AHP) is used to study the relationship between drinking
water quality and landform classes in south of Firozabad, east of Fars province, Iran. For determination
of drinking water quality, parameters of calcium (Ca), chlorine (Cl), magnesium (Mg), thorium (TH),
sodium (Na), electrical conductivity (EC), sulfate ($So_4$) and total dissolved solids (TDS) were used. It
was found that 8.29% of the study area have low water quality; 64.01%, moderate; 23.33%, high; and
very high, 4.38%. Areas with suitable drinking water quality are located in parts of the southeast and
southwest parts of the study area. The relationship between landform class and drinking water quality
show that drinking water quality is high in the stream, valleys, upland drainages and local ridge classes,
and low in the plain small and midslope classes.
**Keywords**: Drinking water quality, fuzzy AHP method, GIS, landform, south of Firozabad.

**1. Introduction**
Landform characteristics can affect the direction of water movement and water quality. Hence, in the
different landforms, there is different water quality (Bise, 2013). To this end, studies on the relationship
between landform classes and water quality have received significant attention. For example, William et
al. (2007) investigated runoff and water quality from three soil landform units on mancos shale. A survey
of sediment basins in steep, dissected shale up lands indicated that an average of 1.25 Mg/ha/year of
sediment is produced by that landform unit carefully designed and located basin plugs can be used
effectively to trap sediment, water, and salt from dissected shale uplands. Mehdi et al. (2012) determined
agricultural land use scenarios for modelling future water quality. The results showed that there is
relationship between types of land use and water quality. The impact of land use on water quality was
evaluated by Huang et al. (2013). The results indicated that there was significant negative correlation
between forest land and grassland and the water pollution, and the built-up area had negative impacts on
the water quality, while the influence of the cultivated land on the water quality was very complex.



In addition, different algorithms have been employed for the determination of water quality. Yonas (2012)
developed a complementary modeling framework to handle systematic error in physically based
groundwater flow model applications that used data-driven models of the errors during the calibration
phase. The effectiveness of four error-correcting data-driven models, namely, artificial neural networks
(ANN), support vector machines (SVM), decision trees (DT) and instance based weighting (IBW) was
examined for forecasting head prediction errors, and subsequently updating the head predictions at
existing and proposed observation wells. Rule based modeling (Manoucher, 2010) was used for spatial
prediction of groundwater quality in Beaufort West, in the Karoo region of South Africa. The
groundwater quality data from about 100 bore wells with a 30 years span collected between 1970 and
2007 was used. The variables used in the analyses included chemicals such as chloride, sulphate,
magnesium, sodium, phosphates and calcium. These were used as predictors for groundwater quality and
electrical conductivity. Aliabadi and Soltanifard (2014) used fuzzy inference for determination of impact
of water and soil electrical conductivity and calcium carbonate on wheat crop using. The inference
system estimated the performance using soil EC, water EC and calcium carbonate in the soil as input
parameters, and also analyzed them.

The aim of this study is the determination of the relationship between landform classes and drinking
water quality in south Firozabad, Iran. In this study, drinking water quality is evaluated using
parameters of calcium (Ca), chlorine (Cl), magnesium (Mg), thorium (TH), sodium (Na), electrical
conductivity (EC), sulfate ($So_4$) and total dissolved solids (TDS). It is proposed that the most
appropriate method to prepare drinking water quality maps is fuzzy analytic hierarchy process (AHP)
in a geographic information system (GIS) environment. It is expected that the determination of the
relationship between landform classes and drinking water quality will allow for the prediction of
drinking water quality based on landform classes. The methodology employed in this study is
summarized in Figure 1.





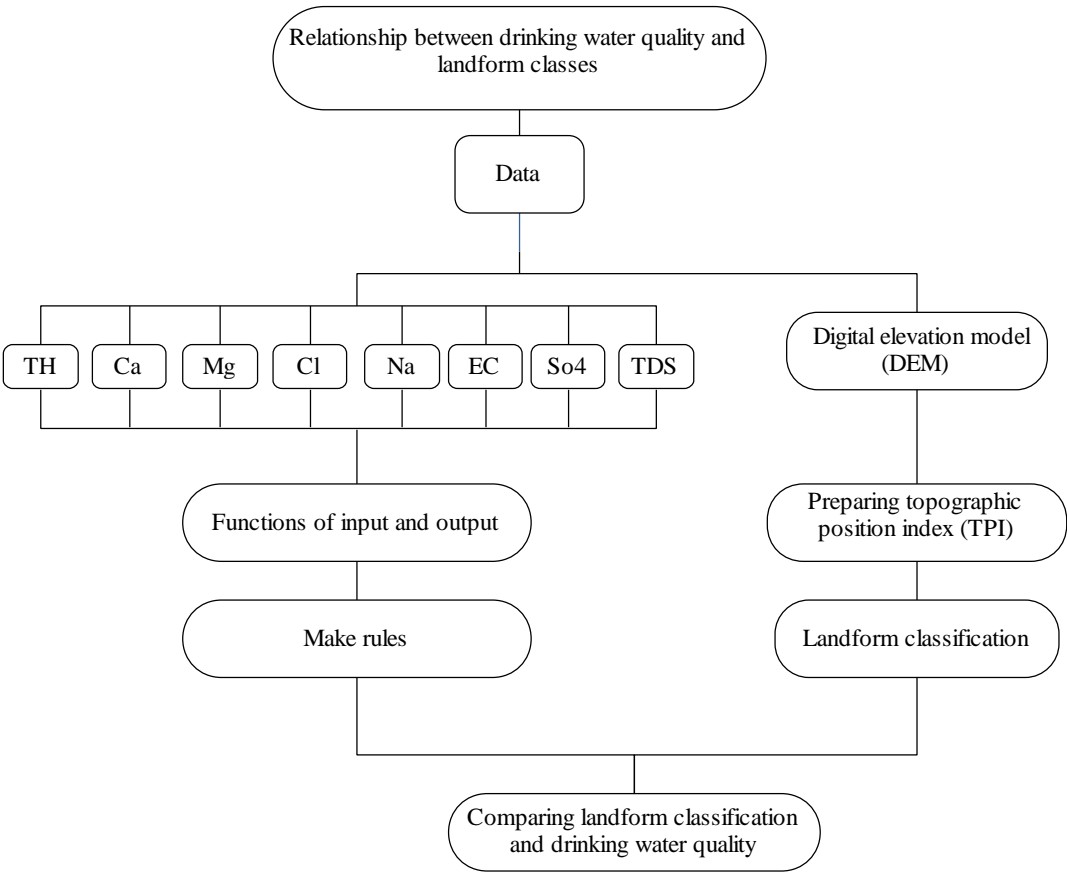

Figure 1. Flowchart for the methodology used in this study to determine the relationship between drinking

water quality and landform classes.


2. Material and method
2.1. Case study
This study was carried out in south of Firozabad, east of Fars Province, Iran. It has an area of 722.91 km$^2$,
and is located between longitude of N 28° 36´- 28° 57´ and latitude  of E 52° 16´ to 52° 46´ (Figure 2).
The altitude of the study area ranges from the lowest of 1,134 m to the highest of 2,885 m. The study area
is abundantly watered by springs and the perennial Firozabad river. The main agricultural produce
consists of grain, fruit, and vegetables, while the partly wooded mountains are used for pasture (Ebn al-
Balkr, 1912; Sharifi-Rad, 2014). The assessment  of land suitability for agricultural production in the
region is vital, which should consider environmental factors and human conditions.

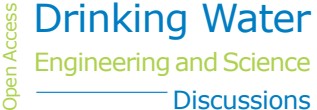


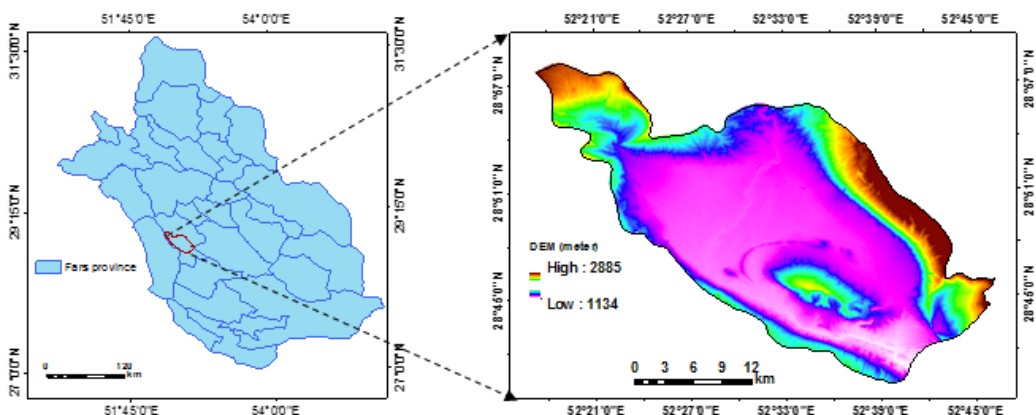

Figure 2. Location of the study area (digital elevation model (DEM) with spatial resolution of 30 m)
(Source: http://earthexplorer.usgs.gov).

One of these important factors is drinking water quality in the study area. In order to predict the
variability of drinking water quality, calcium (Ca), chlorine (Cl), magnesium (Mg), thorium (TH),
sodium (Na), electrical conductivity (EC), sulfate (So$_4$), total dissolved solids (TDS) were prepared
(Table 1) (Fars Regional Water Authority).

Table 1. Descriptive statistics of the parameters for evaluation of water quality (Fars Regional Water
Authority).

| Parameters | Unit | Minimum | Maximum | mean | Stdv dev. |
|---|---|---|---|---|---|
| Calcium (Ca) | mg/l | 0 | 596 | 195 | 89 |
| Chlorine (Cl) | mg/l | 25 | 437 | 84 | 45 |
| Sodium (Na) | mg/l | 0 | 458 | 51 | 45 |
| Electrical, conductivity (EC) | ds/m | 0.39 | 1.75 | 0.71 | 0.15 |
| Magnesium (Mg) | mg/l | 0 | 569 | 182 | 80 |
| Sulfate (So$_4$) | mg/l | 0 | 584 | 137 | 73 |
| Thorium (TH) | mg/l | 0 | 473 | 180 | 77 |
| Total Dissolved Solids (TDS) | mg/l | 0 | 954 | 295 | 117 |






## 2.2. Ordinary Kriging (OK)


The input parameters for determination of drinking water quality are Ca, Cl, Mg, TH, Na, EC, $So_4$ and TDS.
Interpolation maps of these parameters are prepared using ordinary kriging (OK). The presence of a spatial
structure where observations close to each other are more alike than those that are far apart (spatial
autocorrelation) is a prerequisite to the application of geostatistics (Goovaerts, 1999). The experimental
variogram measures the average degree of dissimilarity between unsampled values and a nearby data
value, and thus, can depict autocorrelation at various distances. The value of the experimental variogram
for a separation distance of $h$ (referred to as the lag) is half the average squared difference between the
value at $z(x_i)$ and the value at $z(x_i + h)$: (Oliver, 1990):

$$\bar{\gamma}(h) = \frac{1}{2N(h)} \sum_{i=1}^{N(h)} \left[ z(x_i) - z(x_i + h) \right]^2$$

(1)


where $N$ is the number of pairs of sample points $z(x_i)$ and $z(x_i+h)$ separated by distance $h$ and $\bar{\gamma}(h)$ is the
semivariogram. From the analysis of the experimental variogram, a suitable model is then fitted, usually
by weighted least squares and four parameters; sill, range, nugget and anisotropy. Sill refers to the
variance value at which the curve reaches the plateau sill. The total separation distance from the lowest
variance to the sill is known as range. Semivariogram modeling is a key step between spatial description
and spatial prediction. The main application of kriging is the prediction of attribute values at unsampled
locations. There are several models for semivariogram graphs. Figure 3 shows the general shapes and
equations of the mathematical models used to describe the semivariance (McBratney and Webster, 1986).

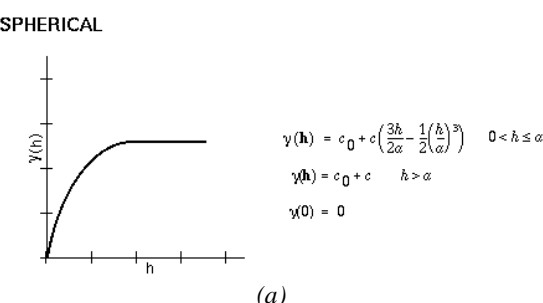

*(a)*



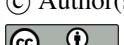

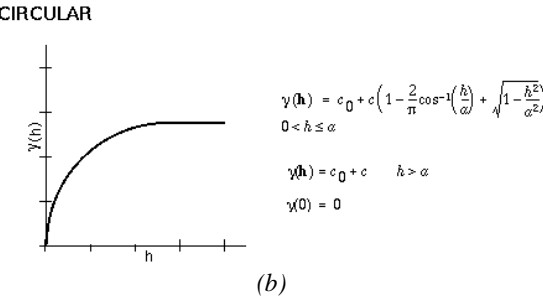

*(b)*

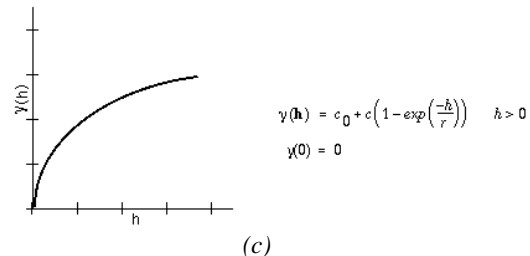

*(c)*

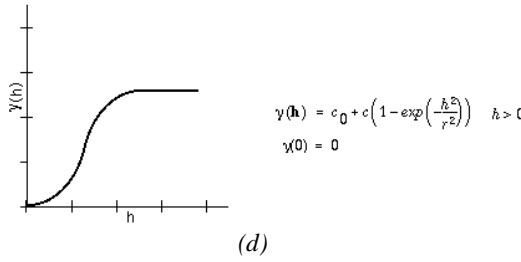

*(d)*

Figure 3. Semivariogram graphs: (a) Spherical  (b) Circular  (c) Exponential  (d) Gaussian

In order to compare, the different interpolation techniques, we examined the difference between known and predicted data using root mean squared error (RMSE) (Eq. (2))

$$\text{RMSE} = \sqrt{\frac{1}{N}\sum_{i=1}^{N}\{z(x_i) - \hat{z}(x_i)\}^2} \qquad (2)$$

where $\hat{z}(x_i)$ is the predicted value, $z(x_i)$ is the observed (known) value, and $N$ is the number of values in the dataset (Johnston et al., 2001).

**2.3 Fuzzy AHP**
**Fuzzy classification**
Fuzzy logic was initially developed by Zadeh (1965) as a generalization of classic logic. He defined a
fuzzy set by memberships function from properties of objects. A membership function assigns to each
object a grade ranging between 0 and 1 .The value 0 means that $x$ is not a member of the fuzzy set, while
the value 1 means that $x$ is a full member of the fuzzy set. Traditionally, thematic maps represent discrete
attributes based on Boolean memberships, such as polygons, lines and points. Mathematically, a fuzzy set
can be defined as following (Mc Bratney and Odeh, 1997):

(3)

where $\mu_A$ is the membership function (MF) that defines the grade of membership of $x$ in fuzzy set $A$. MF
takes values between and including 1 and 0 for all $A$, with $\mu_A = 0$ meaning that $x$ does not belong to $A$ and
$\mu_A = 1$ meaning that it belongs completely to $A$. Alternatively, $0 < \mu_A(x) < 1$ implies that $x$ belongs in a
certain degree to $A$. If $X=\{x_1, x_2, ...., x_n\}$ the previous equation can be written as following (McBratney and
Odeh, 1997):
$A = \{[x_1, \mu_A(x_1)] + [x_2, \mu_A(x_2)] + ...... + [x_n, \mu_A(x_n)]\}$                    (4)
In simple terms, Equations (3) and (4) mean that for every $x$ that belongs to the set $X$, there is a
membership function that describes the degree of ownership of $x$ in $A$.

The development of GIS has contributed to facilitate the mapping of drinking water quality using both
Boolean and fuzzy methods. For each of parameters, the following function was used (Shobha et al.,

2013):

$\mu_A(X) = f(x) = \begin{cases} 1 & x \le a \\ b-x/b-a & a \prec x \prec b \\ 0 & x \ge b \end{cases}$                    (5)
In order to define the fuzzy rules, the drinking water quality standards in Table 2 were used.





Table 2. Drinking water quality standards (WHO) (Shobha et al., 2013)

| Parameters | Permissible limit (mg/liter) |
|---|---|
| Calcium (Ca) | 200 |
| Chlorine (Cl) | 200 |
| Magnesium (Mg) | 150 |
| Thorium (TH) | 500 |
| Sodium (Na) | 200 |
| Electrical conductivity (EC) | 3000 |
| Sulfate (So$_4$) | 200 |
| Total Dissolved Solids (TDS) | 500 |

**Analytic hierarchy process (AHP)**
AHP is a structured technique for organizing and analyzing complex decisions. This method is based on a
pair-wise comparison matrix. The matrix is called consistent if the transitivity (Equation (6)) and
reciprocity (Equation (7)) rules are respected:

$a_{ij} = a_{ik} \cdot a_{kj}$          (6)
$a_{ij} = 1 / a_{ji}$          (7)

where $i$, $j$ and $k$ are any alternatives of the matrix.

In a consistent matrix (Equation (8)), all the comparisons $a_{ij}$ obey the equality $a_{ij} = p_i/p_j$ , where $p_i$ is the
priority of the alternative $i$. When the matrix contains inconsistencies, two approaches can be applied:
$$\begin{vmatrix} P_1/P_1 & \dots & P_1/P_j & \dots & P_1/P_n \\ \dots & 1 & \dots & \dots & \dots \\ P_i/P_1 & \dots & 1 & \dots & P_i/P_n \\ \dots & \dots & \dots & 1 & \dots \\ P_n/P_1 & \dots & P_n/P_j & \dots & P_n/P_n \end{vmatrix}$$
         (8)

In this method, pair-wise comparisons are considered as input, while relative weights are considered as
outputs. The average of each row of the pair-wise comparison matrix is calculated and these average
values indicate relative weights of compared criteria.





**Combination of fuzzy and AHP methods**


Finally, in order to prepare the drinking water quality map, it is necessary to calculate the convex
combination of the raster values containing the different fuzzy parameters. $A_1, ... A_k$ are fuzzy subclasses
of the defined universe of objects $X$, and $W_l, ... W_k$ are non-negative weights summing up to unity. The
convex combination of $A_1, ... A_k$ is a fuzzy class $A$ (Burrough, 1989), and the weights $W_l, ... W_k$ are
calculated using AHP and fuzzy method parameters that have been calculated in ArcGIS. Equations 9 and
10 show the convex combination.
$$\mu_A = \sum_{j=1}^{k} W_j \times \mu_{A(x)} \qquad x \varepsilon X \tag{9}$$

$$\sum_{j=1}^{k} W_j = 1 \qquad W_j > 0 \tag{10}$$

The Fuzzy AHP approach in this study has been divided into five stages, which are summarized in Figure

4.









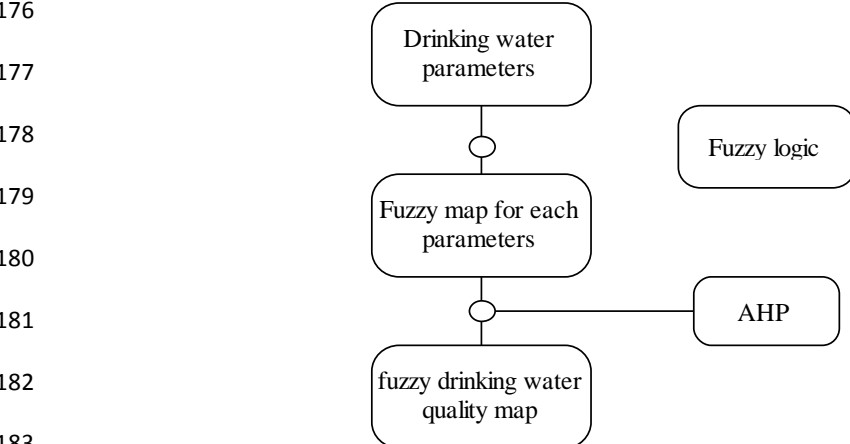

Figure 4. Fuzzy AHP procedure for drinking water quality.

All the model parameters maps are constructed by interpolation between 50 sampling points using the
kriging method. Next, fuzzy logic is applied to create a fuzzy parameter map for each parameter. To
arrive at an integrated evaluation using drinking water quality classes, the fuzzy parameter maps were
aggregated into a drinking water quality map following a weighted summation using AHP.



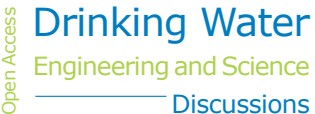


**2.4. Landform Classification Using Topographic Position Index (TPI)**

TPI (Weiss, 2006) compares the elevation of each cell in a DEM to the mean elevation of a specified
neighborhood around that cell. Positive and negative TPI values represent locations that are higher and lower
than the average of their surroundings respectively. TPI values near zero are either flat areas (where the slope
is near zero) or areas of constant slope (where the slope of the point is significantly greater than zero) (Weiss
2006).

TPI (Eq. (11)) compares the elevation of each cell in a DEM to the mean elevation of a specified
neighborhood around that cell. Mean elevation is subtracted from the elevation value at the center (Weiss
2006):

$$TPI_i = T_0 - \frac{\sum_{n-1} T_n}{n} \qquad (11)$$
where;
$T_0$ = elevation of the model point under evaluation
$T_n$ = elevation of grid
$n$ = the total number of surrounding points employed in the evaluation.

Combining TPI at small and large scales allows a variety of nested landforms to be distinguished Table 3.

Table 3. Landform classification based on TPI .(Source: Weiss 2006)

| Classes | Description |
|---|---|
| Canyons, deeply incised streams | Small Neighborhood:$T_o \leq$ -1 |
|  | Large Neighborhood:$T_o \leq$ -1 |
| Midslope drainages, shallow valleys | Small Neighborhood:$T_o \leq$ -1 |
|  | Large Neighborhood: -1 $<T_o<$ 1 |
| upland drainages, headwaters | Small Neighborhood:$T_o \leq$ -1 |
|  | Large Neighborhood:$T_o \geq$ 1 |
| U-shaped valleys | Small Neighborhood: -1 $<T_o<$ 1 |
|  | Large Neighborhood:$T_o \leq$ -1 |
| Plains small | Neighborhood: -1 $<T_o<$ 1 |
|  | Large Neighborhood: -1 $<T_o<$ 1 |
|  | Slope $\leq$ 5° |
| Open slopes | Small Neighborhood: -1 $<T_o<$ 1 |
|  | Large Neighborhood: -1 $<T_o<$ 1 |



|  | Slope > 5° |
|---|---|
| Upper slopes, mesas | Small Neighborhood: $-1 < T_o < 1$ |
|  | Large Neighborhood: $T_o \geq 1$ |
| Local ridges/hills in valleys | Small Neighborhood: $T_o \geq 1$ |
|  | Large Neighborhood: $T_o \leq -1$ |
| Midslope ridges, small hills in plains | Small Neighborhood: $T_o \geq 1$ |
|  | Large Neighborhood: $-1 < T_o < 1$ |
| Mountain tops, high ridges | Small Neighborhood: $T_o \geq 1$ |
|  | Large Neighborhood: $T_o \geq 1$ |


## 4. Results and Discussion

### 4.1. Geostatistical analysis

OK was used for the prediction of the drinking water quality parameters (TH, Ca, Mg, Cl, Na, EC, So4
and TDS). In OK, in order to select the best method (Circular, Spherical, Exponential and Gaussian),
measured nugget, partial sill and RMSE were used (Table 4). The RMSE of water parameters from Table
4 shows that the lowest RMSE is the Gaussian method. Furthermore, these results indicate that the
Gaussian model for OK is the best semivariogram model to show the strong spatial dependency for the
water variable.
Table 4. Sampling nugget, partial sill and RMSE of the different interpolated methods for predicted
drinking water quality using MLR.

| Methods | Model | Parameter | Nugget | Partial Sill | RMSE |
|---|---|---|---|---|---|
|  |  | TDS | 0.66 | 0.32 | 0.80 |
|  |  | TH | 0.7 | 0.229 | 0.80 |
|  |  | Ca | 0.71 | 0.20 | 0.92 |
|  | Circular | Mg | 0.70 | 0.36 | 0.61 |
|  |  | Na | 0.63 | 0.45 | 0.90 |
|  |  | Cl | 0.57 | 0.38 | 0.77 |
|  |  | So4 | 0.62 | 0.29 | 0.91 |
|  |  | EC | 0.57 | 0.26 | 0.56 |
| OK |  | Parameter | Nugget | Partial Sill | RMSE |
|  |  | TDS | 0.67 | 0.32 | 0.80 |
|  |  | TH | 0.69 | 0.30 | 0.81 |
|  |  | Ca | 0.72 | 0.20 | 0.92 |
|  | Spherical | Mg | 0.70 | 0.37 | 0.61 |
|  |  | Na | 0.63 | 0.44 | 0.90 |
|  |  | Cl | 0.57 | 0.37 | 0.77 |
|  |  | So4 | 0.62 | 0.30 | 0.91 |
|  |  | EC | 0.55 | 0.28 | 0.56 |
|  |  | Parameter | Nugget | Partial Sill | RMSE |



| | Parameter | Nugget | Partial Sill | RMSE |
|---|---|---|---|---|
| Exponential | TDS | 0.62 | 0.32 | 0.81 |
| | TH | 0.63 | 0.37 | 0.82 |
| | Ca | 0.70 | 0.20 | 0.93 |
| | Mg | 0.69 | 0.36 | 0.62 |
| | Na | 0.63 | 0.45 | 0.91 |
| | Cl | 0.55 | 0.35 | 0.78 |
| | So4 | 0.56 | 0.36 | 0.92 |
| | EC | 0.44 | 0.39 | 0.62 |
| | Parameter | Nugget | Partial Sill | RMSE |
| | TDS | 0.67 | 0.32 | 0.79 |
| | TH | 0.73 | 0.27 | 0.80 |
| | Ca | 0.71 | 0.21 | 0.91 |
| Gaussian | Mg | 0.71 | 0.36 | 0.60 |
| | Na | 0.64 | 0.45 | 0.90 |
| | Cl | 0.57 | 0.39 | 0.76 |
| | So4 | 0.66 | 0.26 | 0.89 |
| | EC | 0.57 | 0.26 | 0.53 |


Each of water parameters map that was predicted by OK is shown in Figure 5. The lowest So4, TDS, Na,
Mg, TH and Ca were 0, while the highest values for the parameters were 589, 954, 458, 569, 473 and 569
mg/l respectively. The lowest values for EC and Cl were 0.39 and 25 mg/l respectively, while the highest
were 1.7 and 437 respectively. In the total, the results showed that expect for Ca and Mg, the other
parameters had high values in the study area.

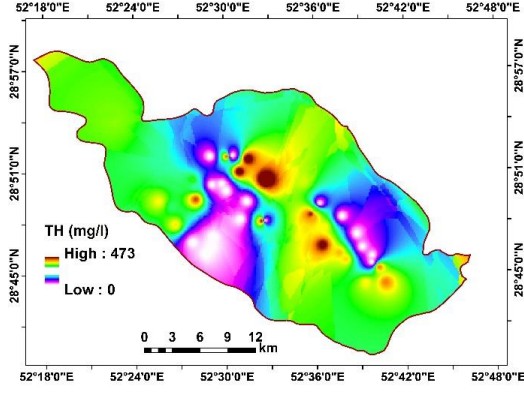

TH                                                    Ca



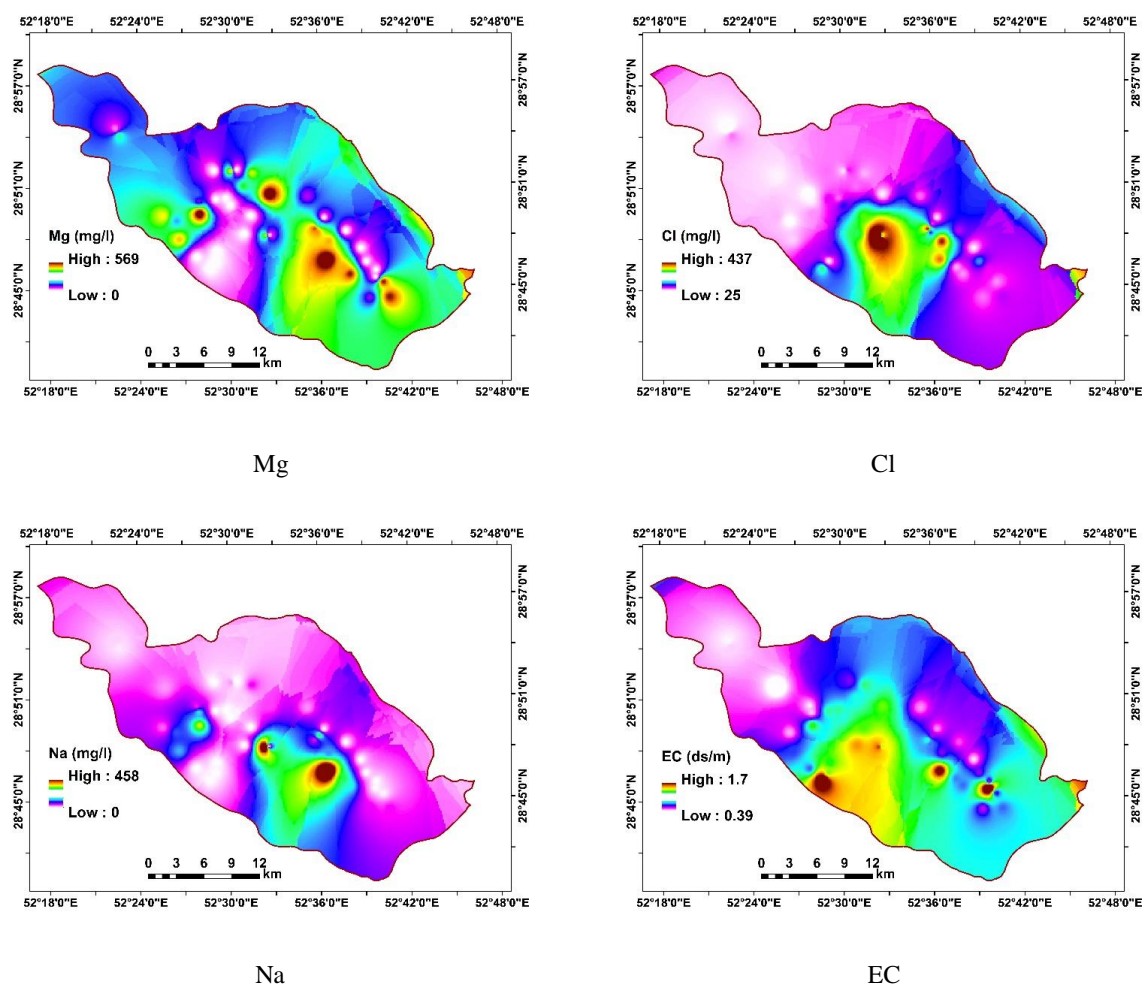

Mg                                          Cl

Na                                          EC



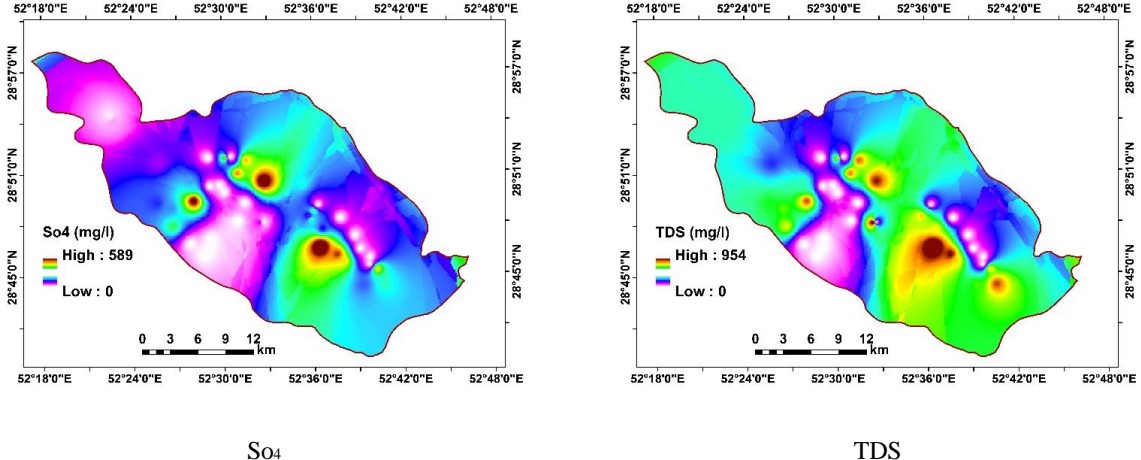

So₄                                              TDS

Figure 5. Interpolated maps of the drinking water quality parameters generated using by OK.


**4.2. Fuzzy method**
The fuzzy maps prepared for the drinking water quality parameters are shown in Figure 6, where MF is
closer to 0 with decreasing drinking water quality, while MF is closer to 1 with increasing drinking water
quality (Soroush et al., 2011). Next, the AHP method was applied on the fuzzy parameter maps. The
pair-wise comparison matrix used for preparation of the weights for each parameter in AHP are given in
Table 5. The drinking water quality map generated using fuzzy-AHP is shown in Figure 7.

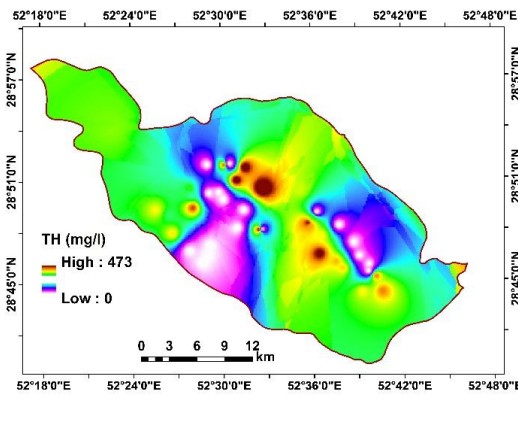
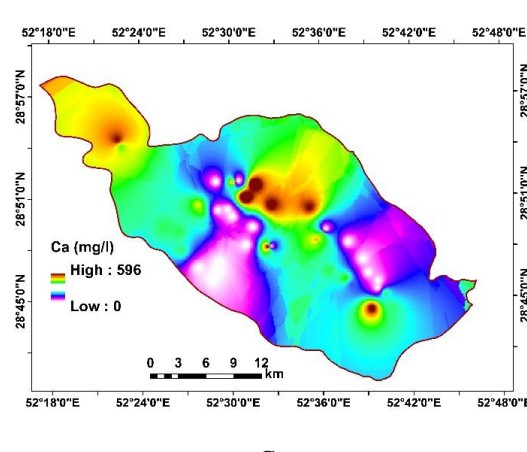

TH                                              Ca





Mg

Cl

Na

EC

So₄

TDS

Figure 6. Fuzzy maps of study area for the drinking water quality parameters.

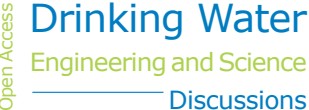



Table 5. Pair-wise comparison matrix for drinking water quality.

| parameters | Ca | Cl | Na | EC | Mg | So4 | TH | TDS | Weight |
|---|---|---|---|---|---|---|---|---|---|
| Ca | 1 | 2 | 3 | 4 | 5 | 6 | 7 | 8 | 0.33 |
| Cl | 0.5 | 1 | 2 | 3 | 4 | 5 | 6 | 7 | 0.23 |
| Na | 0.33 | 0.5 | 1 | 2 | 3 | 4 | 5 | 6 | 0.16 |
| EC | 0.25 | 0.33 | 0.5 | 1 | 2 | 3 | 4 | 5 | 0.11 |
| Mg | 0.2 | 0.2 | 0.33 | 0.5 | 1 | 2 | 3 | 4 | 0.07 |
| So4 | 0.16 | 0.16 | 0.2 | 0.33 | 0.5 | 1 | 2 | 3 | 0.05 |
| TH | 0.14 | 0.14 | 0.16 | 0.2 | 0.33 | 0.5 | 1 | 2 | 0.03 |
| TDS | 0.12 | 0.12 | 0.14 | 0.16 | 0.2 | 0.33 | 0.5 | 1 | 0.02 |


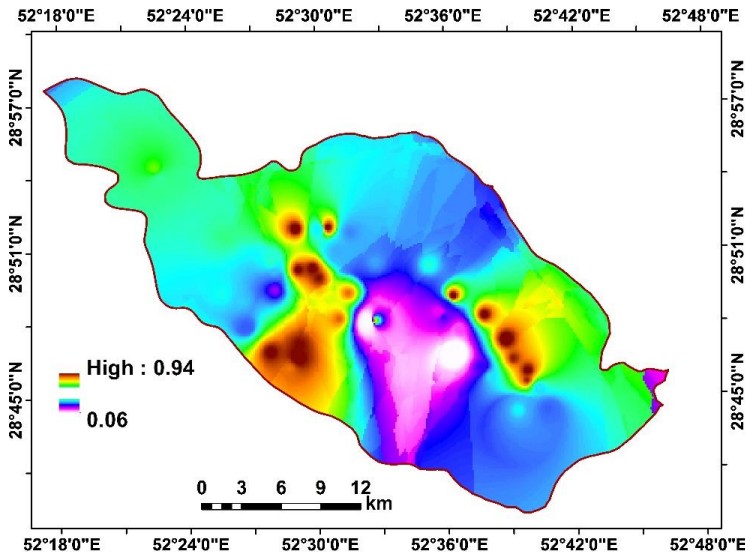


Figure 7. Drinking water quality map generated using fuzzy AHP.


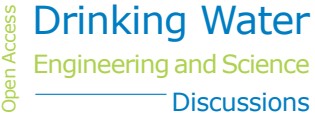


The drinking water quality map is classified into four classes (Mokarram et al., 2010; Shobha et al.,

2013):

➢  Low (not suitable for drinking): < 0.25

➢  Moderate: 0.25 – 0.50

➢  High: 0.50 – 0.75

➢  Very high (suitable for drinking): > 0.75


The results of the classification are shown in Table 6. It is found that areas with suitable drinking water
quality are located in the southeast and southwest parts of the study area (Figure 7).

Table 6. Areas of the drinking water classes.

| Class | Area | |
|---|---|---|
| | (%) | (km$^2$) |
| Low | 8.29 | 59.90 |
| Moderate | 64.01 | 462.72 |
| High | 23.33 | 168.65 |
| Very high | 4.38 | 31.64 |




**4.3. Landform classification**
In order to determine the relationship between landform classification and drinking water quality, a
landform classification map for the study area was prepared using TPI. The TPI maps generated using
small (3 cells) and large (45 cells) neighborhoods are shown in Figure 8. TPI is between -144 to 147 and -
287 to 492 for the small and large neighborhoods respectively. The landform maps generated based on the
TPI values are shown in Figure 10. The classification has ten classes; high ridges, midslope ridges, upland
drainage, upper slopes, open slopes, plains, valleys, local ridges, midslope drainage and streams (Figure
9). The areas of the landform classes are shown in Figure 10. It is observed that the largest landform is
streams, while the smallest is plains.






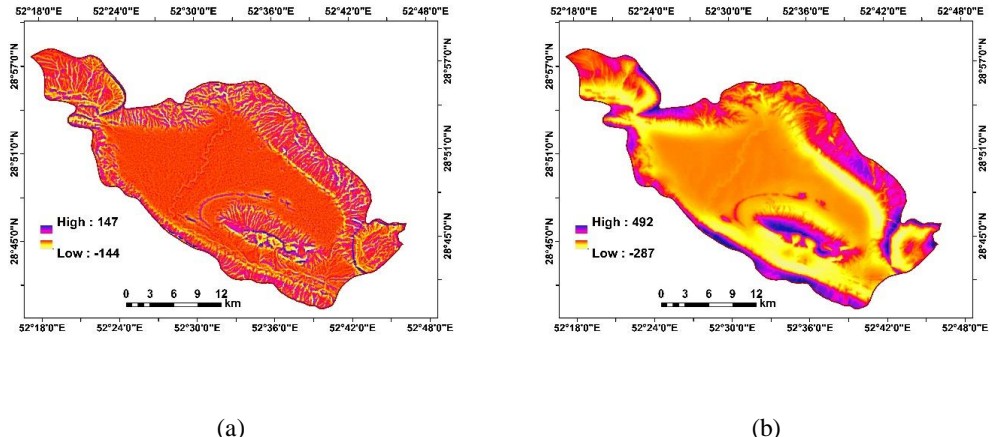

(a)                                                                 (b)


Figure 8. TPI maps generated using (a) small (3 cells) and (b) large (45 cells) neighborhoods.


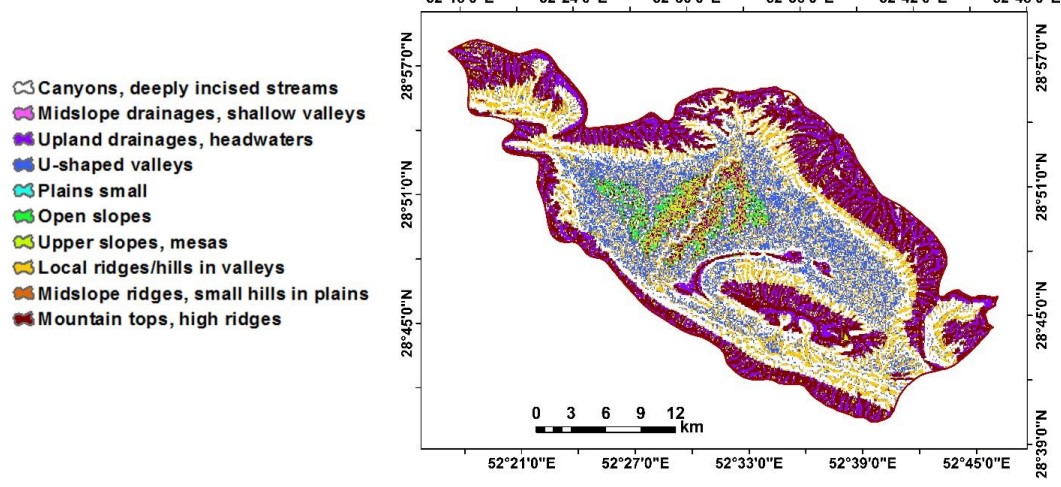


Figure 9. Landform classification using the TPI method.

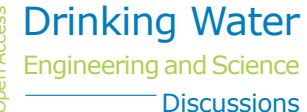


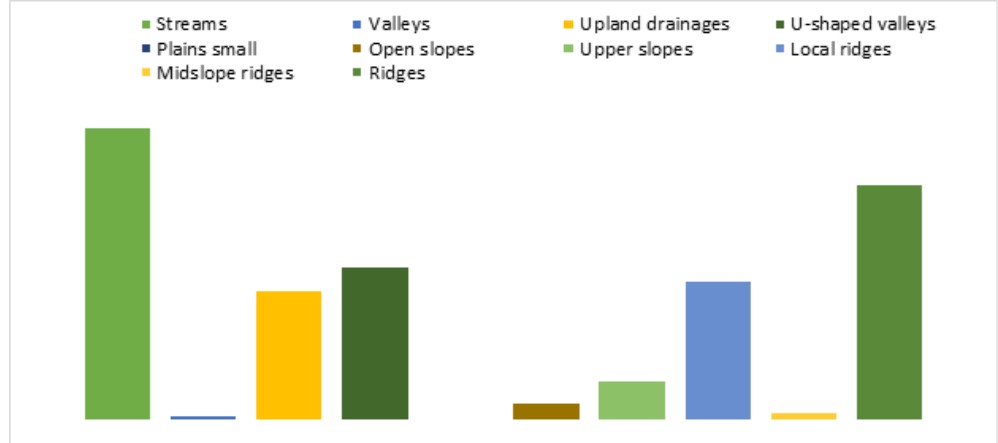

278                    Figure 10. Areas of the landform classes.


The relationship between drinking water quality and landform classes were determined (Figure 11). It is
found that drinking water quality is high for streams, valleys, upland drainages and local ridge classes,
while the lowest drinking water quality is in the plain small and midslope classes. The characteristics of
landform classes, such as slope and geology, determine the drinking water quality. For example, in the
plain small class, due to the low slope, there are ample opportunities for high drinking water quality
(Christiansen, 2004). Therefore, landform maps can be used to predict drinking water quality without
water sampling and analysis in the laboratory.














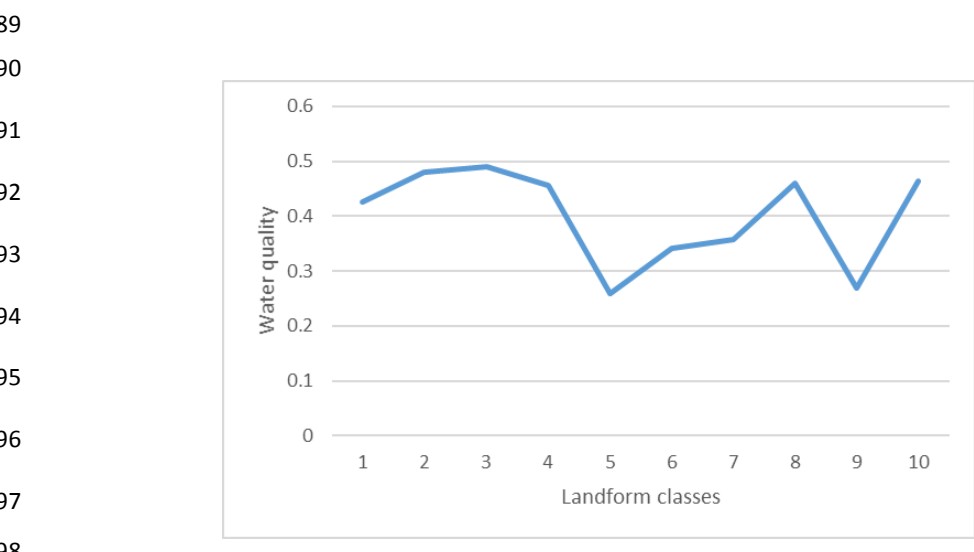

Figure 11. Relationship between drinking water quality and landform classes.



**5. Conclusions**

In this study, fuzzy AHP was used to study the relationship between drinking water quality and landform classes in south of Firozabad. It was found that 8.29% of the study area had low water quality; 64.01%, moderate; 23.33%, high; and 4.38%, very high. The lands suitable for drinking water are located in the southeast and southwest parts of the study area. The relationship between landform class and drinking water quality show that drinking water quality is high in the stream, valleys, upland drainages and local ridge classes, while the lowest drinking water quality is in the plain small and midslope classes.

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
