# Peer review of "Investigation of the relationship between drinking water quality and landform classes using fuzzy AHP(case study: south of Firozabad, east of Fars province, Iran)"

_Drinking Water Engineering and Science, 2016_

## Referee Comment (RC1) · S. K. ROY (Referee) · 18 Jul 2016

Study the relationship between drinking water quality and landform classes is an excellent approach.

Minor comments that may be incorporated:

1. Line 16: For determination of drinking water quality, parameters considered are calcium (Ca), chlorine (Cl), magnesium (Mg), thorium (TH), sodium (Na), electrical conductivity (EC), sulfate (So4) and total dissolved solids (TDS). Dissolved micro organisms are also important. Author could include their view of not including the same

in drinking water quality determination.

2. Line 23: Add one line on how this study of relationship between landform class and drinking water quality will help.

3. line 59- 61 :Please add few sentence on justification for the text "It is proposed that the most appropriate method to prepare drinking water quality maps is fuzzy analytic hierarchy process (AHP) in a geographic information system (GIS) environment."

4. line 61-63 : Pl. provide few sentences on justification "It is expected that the determination of the relationship between landform classes and drinking water quality will allow for the prediction of drinking water quality based on landform classes."

5. Line 166-168 : "... it is necessary to calculate the convex..." . Pl. put reference and few line explanation on why it is necessary.

6. Line 308: Please add benefit of your study and future work that you suggest.
* * *

---

## Author Comment (AC1) · 29 Jul 2016

The comment was uploaded in the form of a supplement:
http://www.drink-water-eng-sci-discuss.net/dwes-2016-3/dwes-2016-3-AC1-supplement.zip

---

## Referee Comment (RC2) · Anonymous Referee #2 · 28 Aug 2016

1. The authors argue that organic matter concentration is low. As a result, it is ignored in this study. However, OC is typically low in most of source water and it is still a very important factor to consider for drinking water. Low OC can greatly influence the safety of drinking water. The author should change "drinking water quality" to "inorganic components in drinking water" in the title and in the manuscript. 2. The grammatical errors should be throughly checked and corrected.
* * *

---

## Author Comment (AC2) · 30 Aug 2016

**Author's Response**

**Dear Editor**

1) Change "drinking water quality" to "inorganic components in drinking water" in the title and in the manuscript.
We changed it in all of the text.

2) The grammatical errors should be throughly checked and corrected.
It was corrected